# Precise synthetic control of exclusive ligand effect boosts oxygen reduction catalysis

Lu Tao[1,2,4], Kai Wang [1,4], Fan Lv[1], Hongtian Mi[2], Fangxu Lin[1], Heng Luo[1], Hongyu Guo[1], Qinghua Zhang[3], Lin Gu [3], Mingchuan Luo[1] & Shaojun Guo [1] ✉

Ligand effect, induced by charge transfer between catalytic surface and substrate in core/shell structure, was widely proved to benefit Pt-catalyzed oxygen reduction reaction by tuning the position of $d$-band center of Pt theoretically. However, ligand effect is always convoluted by strain effect in real core/shell nanostructure; therefore, it remains experimentally unknown whether and how much the ligand effect solely contributes electrocatalytic activity improvements. Herein, we report precise synthesis of a kind of $Pd_3Ru_1$/Pt core/shell nanoplates with exclusive ligand effect for oxygen reduction reaction. Layer-by-layer growth of Pt overlayers onto $Pd_3Ru_1$ nanoplates can guarantee no lattice mismatch between core and shell because the well-designed $Pd_3Ru_1$ has the same lattice parameters as Pt. Electron transfer, due to the exclusive ligand effect, from $Pd_3Ru_1$ to Pt leads to a downshift of $d$-band center of Pt. The optimal $Pd_3Ru_1$/$Pt_{1-2L}$ nanoplates achieve excellent activity and stability for oxygen reduction reaction in alkaline/acid electrolyte.

Oxygen reduction reaction (ORR), as the primary cathodic reaction in clean energy conversion/storage systems such as fuel cells and metal-air batteries, has been of great interest in recent few decades[1,2]. The ORR kinetic barriers and high cost of used Pt electrocatalysts are the major bottlenecks in the development of such sustainable energy devices[3,4]. Tremendous research efforts have been devoted to developing low-Pt electrocatalysts and unraveling the ORR mechanism for accelerating ORR[5–7]. It is well documented that the energy of the $d$-band center of the metal catalyst correlates to the adsorption energy, activation energy, and dissociation energy of small molecules[8]. Based on the acknowledged $d$-band center theory, the position of the $d$-band center of Pt determines the adsorption strength of the oxygenated intermediate on Pt active sites[9,10]. The core/shell structural engineering achieved by the deposition of a few Pt atomic layers, preferably monolayer, on a heterogeneous substrate, has been regarded as an effective strategy to tune the $d$-band center of surface Pt and optimize the adsorption strength based on the well-known strain and/ or ligand effects[11–13]. In terms of the M/Pt core/shell models, strain effect on electrocatalytic activity can be well engineered by tuning the thickness of Pt shells or atomic composition, and the correlation of

geometric strain, $d$-band center, adsorption strength, and catalytic activity of Pt has been studied[14,15]. However, short-range ligand effect, induced by charge transfer between surface Pt and sublayer heteroatoms in the core/shell nanostructures, usually has to be mixed with the strain effect because Pt shell and heterogeneous substrates always differ in lattice parameter[16–18]. This inevitable case in the core/shell nanostructures makes it very difficult to determine and quantify experimentally whether the exclusive ligand effect can greatly boost the intrinsic activity of Pt for ORR, though it is a basic assumption in theoretical calculations.

Herein, we report the precise synthesis of a class of $Pd_3Ru_1$ nanoplates (NPs) with the same lattice parameters as pure Pt for the growth of Pt atomic overlayers for well-revealing exclusive ligand effect for greatly enhancing the ORR electrocatalysis experimentally and computationally. X-ray absorption near-edge structure spectra, extended X-ray absorption fine structure spectra, X-ray photoelectron spectra results, and density functional theory (DFT) calculations reveal that the ligand effect caused by electronic donation from $Pd_3Ru_1$ to Pt can result in an increase of electron density in the $5d$ orbitals of Pt, thus the downshift of $d$-band center of Pt compared to

[1]School of Materials Science and Engineering, Peking University, Beijing 100871, China. [2]School of Materials Science and Engineering, University of Science and Technology Beijing, Beijing 100083, China. [3]Beijing National Laboratory for Condensed Matter and Institute of Physics, Chinese Academy of Sciences, Beijing 100190, China. [4]These authors contributed equally: Lu Tao, Kai Wang. ✉e-mail: guosj@pku.edu.cn

the bulk Pt. The strong exclusive ligand effect from $Pd_3Ru_1$ core to monolayer Pt shell in $Pd_3Ru_1/Pt_{1-2L}$ NPs makes them display remarkable mass activity (MA) of 10.3/4.59 amperes per milligram of Pt at 0.90 V *versus* reversible hydrogen electrode towards ORR in alkaline/acid electrolyte, 51.5/25.5 times higher than those of commercial Pt/C catalyst. Furthermore, $Pd_3Ru_1/Pt_{1-2L}$ NPs can maintain 78.2%/52.1% of MA over 30,000 potential cycles in alkaline/acid electrolyte.

## Results

### Synthesis and structural characterizations

The $Pd_xRu_{1-x}$ NPs were synthesized by using palladium diacetylacetonate $(Pd(acac)_2)$ and dodecacarbonyl triruthenium $(Ru_3(CO)_{12})$ as the metal precursors, oleylamine as the solvent and surfactant, and

ascorbic acid as the reducing agent (details in Methods and Supplementary Table 1). The lattice parameters of $Pd_xRu_{1-x}$ NPs could be adjusted by changing the elemental proportion of Pd to Ru (Supplementary Figs. 1 and 2). The crystal type and lattice spacings of $Pd_3Ru_1$ were found to be consistent with those of Pt. The transmission electron microscopy (TEM) images (Supplementary Fig. 3) and high-angle annular dark-field scanning TEM (HAADF-STEM) image (Fig. 1a) show that the as-synthesized $Pd_3Ru_1$ NPs have the hexagonal morphology with the lateral size of 10–20 nm. High-resolution HAADF-STEM results (Fig. 1b, c) reveal that the $Pd_3Ru_1$ NPs have an average thickness of 1.35 nm (equivalent to about 6 atomic layers thick, counted by green lines in Fig. 1b). The corresponding fast Fourier transform (FFT) patterns, taken from the green dashed regions in Fig. 1b, c, depict the typical characteristic diffraction pattern of [011] zone axes and [111]

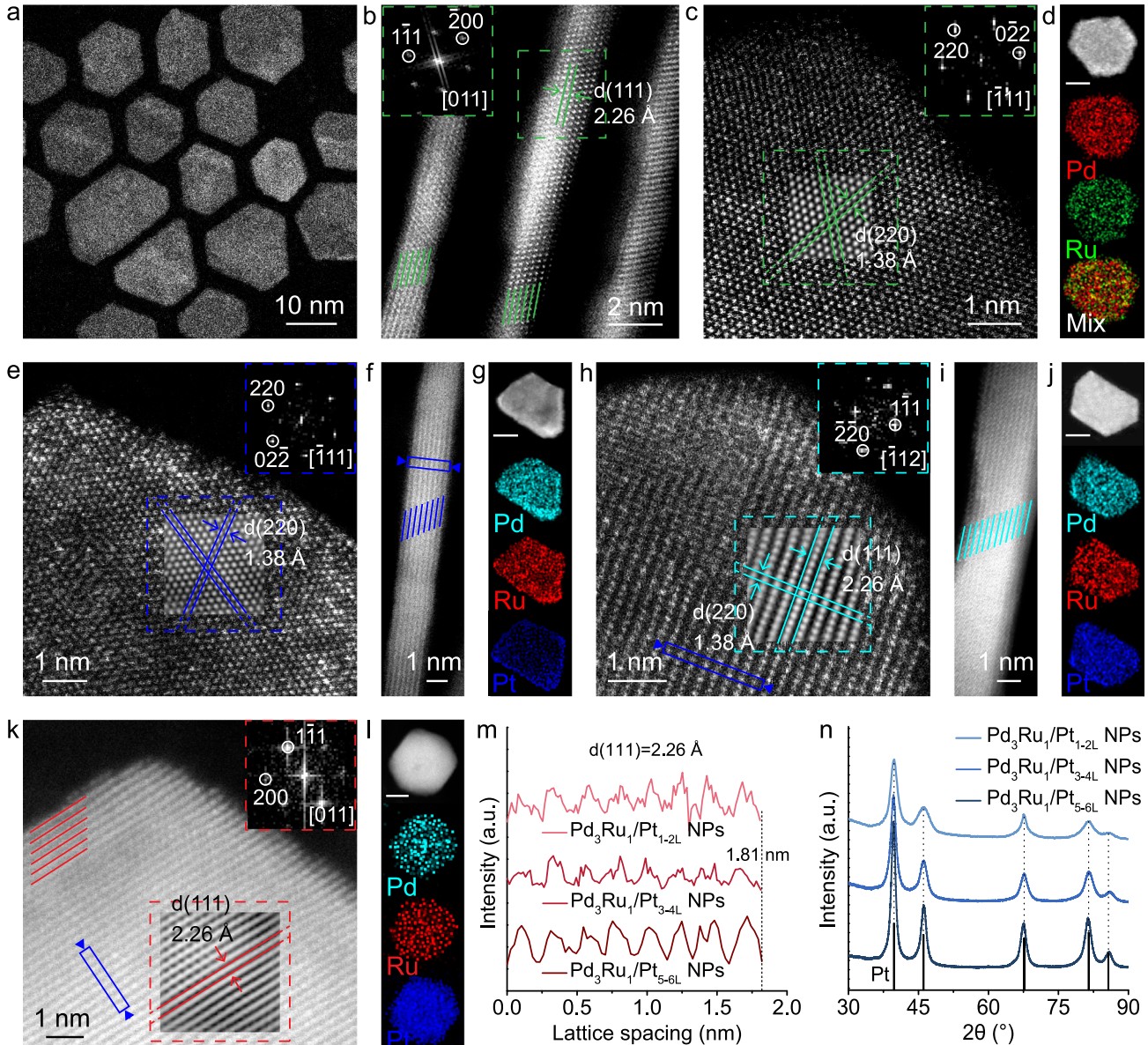

**Fig. 1 | Structural analysis for $Pd_3Ru_1$ NPs and $Pd_3Ru_1/Pt_{nL}$ NPs. a** HAADF-STEM image of $Pd_3Ru_1$ NPs. **b, c** Atomic-resolution HAADF-STEM images of $Pd_3Ru_1$ NP with the corresponding FFT pattern taken from the green dashed area in (**b, c**), respectively. **d** EDS elemental mapping images of $Pd_3Ru_1$ NP. **e, f** Atomic-resolution HAADF-STEM images of $Pd_3Ru_1/Pt_{1-2L}$ NPs with the corresponding FFT pattern taken from the blue dashed square in **e**. **g** EDS elemental mapping images of $Pd_3Ru_1/Pt_{1-2L}$ NP. **h, i** Atomic-resolution HAADF-STEM images of $Pd_3Ru_1/Pt_{3-4L}$ NPs with the corresponding FFT pattern taken from the cyan dashed square in **h**. **j** EDS elemental mapping images of $Pd_3Ru_1/Pt_{3-4L}$ NP. **k** Atomic-resolution HAADF-STEM image of $Pd_3Ru_1/Pt_{5-6L}$ NP with the corresponding FFT pattern taken from the red dashed square in (**k**). **l** EDS elemental mapping images of $Pd_3Ru_1/Pt_{5-6L}$ NP. **m** Integrated pixel intensities of the $Pd_3Ru_1/Pt_{nL}$ phases taken from the blue solid rectangles in (**f, h, k**), respectively. **n** PXRD patterns of $Pd_3Ru_1/Pt_{nL}$ NPs. PDF card: Pt 04-0802. The scale bars in (**d, g, j, l**): 5 nm.

zone axes of the face-centered cubic (*fcc*) structure, respectively. Accordingly, the lattice spacings of (111) plane (Fig. 1b) and (220) plane (Fig. 1c), are calculated to be 2.26 Å and 1.38 Å, respectively, in consistent with those of Pt (111) and Pt (220) (Supplementary Fig. 4). The energy-dispersive X-ray spectroscopy (EDS) elemental mappings (Fig. 1d) of one typical $Pd_3Ru_1$ NP and corresponding line-scan profile (Supplementary Fig. 5a) across the NP reveal the entire-overlapping of Pd and Ru elemental signals in the NP, demonstrating the absence of composition segregation in the $Pd_3Ru_1$ phase. The element ratio of Pd to Ru is determined to be 74.7 to 25.3, in consistent with the inductively coupled plasma optical emission spectrometry (ICP-OES) result (Supplementary Table 2). The powder X-ray diffraction (PXRD) pattern (Supplementary Fig. 5b) of $Pd_3Ru_1$ NPs shows five prominent diffraction peaks located at 39.8°, 46.2°, 67.5°, 81.3°, and 85.7°, assigned to the (111), (200), (220), (311), and (222) planes of *fcc* crystal phase, respectively, without observable shifts compared with those of commercial carbon-supported platinum nanoparticles (Pt/C), further confirming that $Pd_3Ru_1$ NPs have the same crystal structure and lattice parameters as Pt.

The deposition of atomic Pt overlayers on the obtained $Pd_3Ru_1$ NPs was performed in a layer-by-layer fashion through a wet-chemical approach (details in Methods). The well-defined $Pd_3Ru_1/Pt_{nL}$ (n = 1–6) core/shell NPs could be achieved by controlling the amount of Pt precursor (Supplementary Fig. 6 and Table 3). The structural parameters of Pt shells were identified from the atomic stacking sequence and corresponding FFT pattern. The simulated high-resolution HAADF-STEM images of the surface atomic arrangement on the Pt shells with the corresponding FFT pattern are taken from the dashed squares in Fig. 1e, h, k, demonstrate that the corresponding $Pt_{nL}$ shells are in accordance with the (111), (112), and (011)-oriented *fcc* structure, respectively. The atomic-resolution HAADF-STEM images (Fig. 1f, i) of one side view on the obtained NP reveals that the total thickness of NP increases to 8 atomic layers (counted by blue lines, Fig. 1f) and 12 atomic layers (counted by cyan lines, Fig. 1i), respectively. Compared with the original thickness of $Pd_3Ru_1$ NP with 6 atomic layers (Fig. 1b), the foreign Pt shell is determined to be about 1–2 monolayer thick in the Figs. 1f, and 3–4 monolayer thick in the Fig. 1i, respectively. Furthermore, a 6-atom-thick Pt shell (counted by red lines, Fig. 1k) was formed on the $Pd_3Ru_1$ NP with continuous lattice fringes across the interface, indicating the epitaxial growth of Pt overlayers on the $Pd_3Ru_1$ NPs. The EDS elemental mapping analyses further confirm the core/shell structures of $Pd_3Ru_1/Pt_{nL}$ NPs (Fig. 1g, j, l), in which the countable atomic Pt shell and heterogeneous core can be distinguished from the contrast of different elemental signals. The integrated pixel intensities of the (111) lattices taken from various selected areas (boxed by the blue rectangular) of $Pd_3Ru_1/Pt_{1-2L}$ NPs (Fig. 1f), $Pd_3Ru_1/Pt_{3-4L}$ NPs (Fig. 1h) and $Pd_3Ru_1/Pt_{5-6L}$ NPs (Fig. 1k) are shown in Fig. 1m, where all of the average (111) distances are calculated to be 2.26 Å, identical to $Pd_3Ru_1$ (111) (Fig. 1b) and bulk Pt (111) (Supplementary Fig. 4), illustrating inexistence of geometric strain in the primary {111} crystal planes of $Pd_3Ru_1/Pt_{1-6L}$ NPs. The PXRD patterns (Fig. 1n) of various $Pd_3Ru_1/Pt_{1-6L}$ NPs reveal that there are no shifts on five prominent diffraction peaks relative to those of $Pd_3Ru_1$ NPs and commercial Pt/C (Supplementary Fig. 5b), confirming the absence of strain in the $Pt_{1-6L}$ shells. Furthermore, the geometric phase analysis (GPA) on the random $Pd_3Ru_1/Pt_{nL}$ NPs was performed to recheck strain on the Pt shells. The corresponding false-colored GPA map shows almost uniform color throughout the NP, reconfirming a negligible in-plane strain ($\varepsilon_{xx}$) on the Pt shells (Supplementary Fig. 7).

### Investigations of electronic structure and ligand effect

The electronic structure of Pt and the electronic interaction between $Pd_3Ru_1$ core and Pt shell were studied by X-ray absorption spectroscopy (XAS). Figure 2a displays the X-ray absorption near-edge structure (XANES) spectra of Pt $L_3$-edge of various $Pd_3Ru_1/Pt_{nL}$ NPs and Pt

foil. The intensity of Pt $L_3$-edge white line (WL) is regarded as a qualitative indicator of electron density in the $5d$ orbitals of Pt atoms[19]. As shown in Fig. 2a, the peak intensity of various catalysts in the Pt $L_3$-edge WL region follows the order: Pt foil > $Pd_3Ru_1/Pt_{5-6L}$ NPs > $Pd_3Ru_1/Pt_{3-4L}$ NPs > $Pd_3Ru_1/Pt_{1-2L}$ NPs (*inset* of Fig. 2a), indicating that the electron density of Pt shell in the $Pd_3Ru_1/Pt_{nL}$ NPs is successively increased as the thickness of Pt shell decreases.

Besides, the distances of Pt–Pt bonds were quantitatively investigated by Fourier transform extended X-ray absorption fine structure (FT-EXAFS) (Fig. 2b, Supplementary Fig. 8 and Table 4), and the corresponding atomic dispersions and bonding circumstances were displayed in the wavelet transform (WT) of *k* space data (Fig. 2c). It can be seen that the interatomic distance of Pt–Pt in the $Pd_3Ru_1/Pt_{5-6L}$ NPs (2.75 Å), $Pd_3Ru_1/Pt_{3-4L}$ NPs (2.75 Å) and $Pd_3Ru_1/Pt_{1-2L}$ NPs (2.76 Å) is almost identical with that of Pt foil (2.76 Å), which rules out a geometric strain in the Pt shells of $Pd_3Ru_1/Pt_{nL}$ NPs. In consideration of the excluded strain effect, the changed electron density of Pt in the $Pd_3Ru_1/Pt_{nL}$ NPs can be assigned to the electronic effect that electron transfer induces the change of the *d*-band electron density of Pt[20].

The changes in valence states of Pt shells and $Pd_3Ru_1$ core were studied by X-ray photoelectron spectroscopy (XPS). As displayed in Fig. 3a, the Pt $4f$ binding energy of Pt shells in the $Pd_3Ru_1/Pt_{nL}$ NPs exhibits a gradient negative shift with a decreasing thickness of Pt shells whereas the Pd $3d$ and Ru $3p$ binding energies of $Pd_3Ru_1/Pt_{nL}$ NPs show a gradient positive shift with the decreased Pt shells (Supplementary Fig. 9). The increased shifts of Pt $4f$, Pd $3d$ and Ru $3p$ binding energies in the $Pd_3Ru_1/Pt_{nL}$ NPs (from n: 5–6 L, to n: 3–4 L, to n: 1–2 L) represent an enhanced electronic effect that induced by electron transfer from $Pd_3Ru_1$ cores to Pt shells. According to the XPS valence-band spectra results (Fig. 3b), the *d*-band center of $Pd_3Ru_1/Pt_{nL}$ NPs downshifts successively in the order of $Pd_3Ru_1/Pt_{5-6L}$ NPs (−3.702 eV) > $Pd_3Ru_1/Pt_{3-4L}$ NPs (−3.748 eV) > $Pd_3Ru_1/Pt_{1-2L}$ NPs (−3.829 eV), revealing a shift of *d*-band center of $Pt_{nL}$ away from the Fermi level in comparison with Pt/C (−3.698 eV) (Fig. 3c). Therefore, the electronic structure of surface Pt is tuned by the electronic effect from $Pd_3Ru_1$ core, leading to a downshift of Pt *d*-band center with respect to the Fermi level. Such electronic interaction between Pt shells and heterogenous cores (ligand effect) in the $Pd_3Ru_1/Pt_{1-6L}$ configuration is much stronger in the case of monolayer Pt ($Pt_{1-2L}$) and becomes weak ($Pt_{3-4L}$) or even disappears ($Pt_{5-6L}$) with increasing the thickness of Pt shell.

DFT calculation was performed to give a theoretical analysis of the *d*-band center shift and Bader charge of Pt. As shown in the Fig. 3d, the peak of normalized *d*-projected density of states (DOS) of Pt surfaces in different models (Supplementary Fig. 10) demonstrates a smoothing trend from pristine Pt, to $Pt_{6L}$, to $Pt_{4L}$ then to $Pt_{2L}$, indicating that bonding states between atoms become weaker in the presence of thinner Pt layers of $Pd_3Ru_1/Pt_{nL}$ model. The weaker bond interactions, the greater the deviation of the local average of the *d* electron energies of $Pt_{nL}$ from the benchmark value of pristine Pt. Hence, the *d*-band center of respective $Pt_{nL}$ shows up reasonable downshift relative to that of pure Pt (-2.12 eV) away from the Fermi level, as the order of $Pd_3Ru_1/Pt_{6L}$ (−2.37 eV) > $Pd_3Ru_1/Pt_{4L}$ (−2.48 eV) > $Pd_3Ru_1/Pt_{2L}$ (−2.61 eV) (Fig. 3e), which is consistent with the experimental results.

The Bader charge ($\Delta Q$) analysis of $Pt_{nL}$ was further carried out to investigate the average electron transfer between $Pt_{nL}$ and $Pd_3Ru_1$, and the pure Pt without charge transfer ($\Delta Q = 0$) was utilized as a reference. As displayed in the Fig. 3e, the $Pd_3Ru_1/Pt_{2L}$ showcases the greatest number of electrons being transferred; approximately 0.097e contributes from $Pd_3Ru_1$ to $Pt_{2L}$. Likewise, the $Pd_3Ru_1/Pt_{4L}$ and the $Pd_3Ru_1/Pt_{6L}$ exhibit weaker charge transfers of 0.037e and 0.016e from $Pd_3Ru_1$ to $Pt_{4L}$ and $Pt_{6L}$, respectively. This result theoretically confirms that effective charge transfer at the interface are pronounced in the $Pd_3Ru_1/Pt_{2L}$; meanwhile, the thicker of $Pt_{nL}$, the weaker the electronic effect in the $Pd_3Ru_1/Pt_{nL}$.

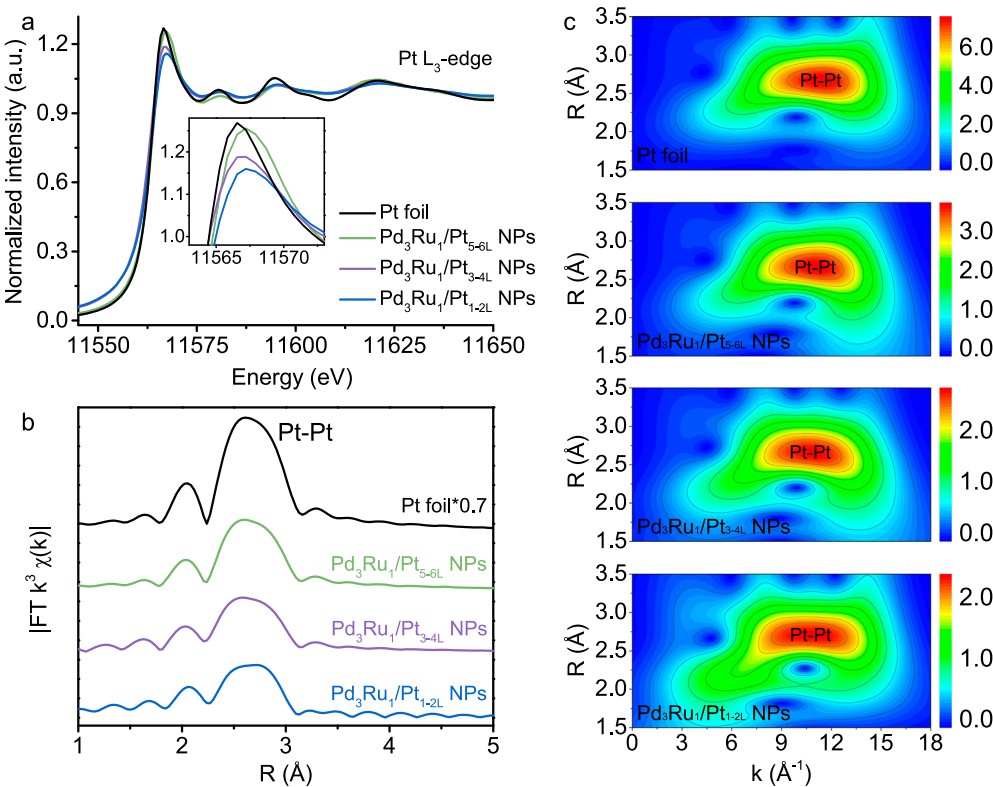

**Fig. 2 | Spectroscopic characterization of Pd₃Ru₁/PtₙL NPs and Pt foil. a** XANES spectra at the Pt $L_3$-edge. *Inset,* enlarged view of WL peak intensity. **b** $k^3$-weighted FT-EXAFS spectra at $R$ space. **c** Pt $L_3$-edge WT analyses.

## Electrocatalytic ORR

The ligand effect in the Pd₃Ru₁/PtₙL NPs catalysts on ORR activity was examined in the O₂-saturated 0.1 M KOH solution. The commercial Pt/C (20.0 wt% of 3 nm-Pt nanoparticles supported on carbon) was used as a benchmark catalyst (Supplementary Fig. 11). The ORR polarization curves of different catalysts at a rotation rate of 1600 rpm and a sweep rate of 20 mV/s at room temperature were recorded in Fig. 4a. The Pd₃Ru₁/Pt₁₋₂L NPs catalyst exhibits the highest half-wave potential ($E_{1/2}$) of 0.945 V, higher than those of Pd₃Ru₁/Pt₃₋₄L NPs (0.900 V), Pd₃Ru₁/Pt₅₋₆L NPs (0.867 V) and commercial Pt/C (0.855 V), suggestive of its most prominent activity toward ORR. Normalized with Pt loadings, the MAs of different catalysts at 0.9 V were calculated to follow by the order of Pd₃Ru₁/Pt₁₋₂L NPs (10.3 A/mgₚₜ) > Pd₃Ru₁/Pt₃₋₄L NPs (1.01 A/mgₚₜ) > Pd₃Ru₁/Pt₅₋₆L NPs (0.31 A/mgₚₜ) > commercial Pt/C (0.20 A/mgₚₜ) (Fig. 4b) whereas normalized with the electrochemical active surface areas (Supplementary Table 5), the specific activities (SA) follow the order of Pd₃Ru₁/Pt₁₋₂L NPs (7.11 mA/cm²) > Pd₃Ru₁/Pt₃₋₄L NPs (1.30 mA/cm²) > Pd₃Ru₁/Pt₅₋₆L NPs (0.71 mA/cm²) > commercial Pt/C (0.31 mA/cm²). As is known, an ideal ORR electrocatalyst should have a weaker binding energy to oxygen relative to Pt (111) by 0.2 eV, and an optimal downshift in the *d*-band center should be 0.2 eV lower than that of Pt[9]. In agreement with the *d*-band model, the Pd₃Ru₁/Pt₁₋₂L NPs, with a downshift in the *d*-band center of 0.131 eV lower than that of Pt (the nearest to optimal value), show the highest ORR activity among different Pd₃Ru₁/PtₙL NPs catalysts, and also are better than most of the reported Pt-based catalysts (Supplementary Table 6).

We also find that the Pd₃Ru₁/Pt₁₋₂L NPs with strong ligand effect show the remarkable electrocatalytic MA of 10.3 A/mgₚₜ and SA of 7.11 mA/cm², 51.5-fold and 22.9-fold higher than those of commercial Pt/C catalysts, whereas the Pd₃Ru₁/Pt₃₋₄L NPs with the weak ligand effect that triggered a slight downshift in the *d*-band center show 5.05-fold and 4.19-fold higher MA and SA than those of commercial Pt/C catalysts, respectively, and the Pd₃Ru₁/Pt₅₋₆L NPs catalyst shows the

comparable ORR activity as that of commercial Pt/C. The above results further reveal that a downshift of the *d*-band center of Pt induced by the ligand effect enhances the ORR catalytic activity.

The ORR electrochemical durability tests were measured in O₂-saturated 0.1 M KOH solution by sweeping the potential cycles between 0.6 and 1.0 V at a scanning rate of 50 mV/s (Supplementary Fig. 12). The changes of MAs at 0.9 V and electrochemically active surface areas (ECSAs) during the tests were summarized in Fig. 4c and d. After 30,000 potential cycles, the MA of the Pd₃Ru₁/Pt₁₋₂L NPs was still as high as 7.83 A/mgₚₜ, 39.2-fold higher than that of the initial commercial Pt/C catalyst, and 87.0-fold higher than that of commercial Pt/C catalyst over 30,000 potential cycles. Furthermore, the Pd₃Ru₁/Pt₃₋₄L NPs and Pd₃Ru₁/Pt₅₋₆L NPs retain 78.2% and 87.1% of the initial MA after 30,000 potential cycles, respectively, much higher than that of commercial Pt/C catalyst (45.0%) (Fig. 4c). As for ECSAs over 30,000 potential cycles (Fig. 4d), there remain high for the Pd₃Ru₁/PtₙL NPs with Pt₁₋₂L, Pt₃₋₄L, and Pt₅₋₆L by maintaining 76.7 %, 78.2%, and 87.2% of their initial values, respectively, superior to Pt/C catalyst with the maintenance of 44.1%. Furthermore, the Pd₃Ru₁/PtₙL NPs show negligible change in their hexagonal morphology (Supplementary Fig. 13) after 30,000 potential cycles; however, the commercial Pt/C catalyst displays severe nanoparticulate aggregation under the same test condition (Supplementary Fig. 14). The high ORR durability of Pd₃Ru₁/PtₙL NPs is believed to be relevant to the electronic interaction between Pd₃Ru₁ and Pt that renders surface Pt atoms less oxophilic and more antioxidative during the catalytic process[16,21,22].

The ligand effect of the Pd₃Ru₁/PtₙL NPs catalysts in ORR activity was further examined in the O₂-saturated 0.1 M HClO₄ solution (Supplementary Fig. 15a). The MAs of different catalysts at 0.9 V follow the order of Pd₃Ru₁/Pt₁₋₂L NPs (4.59 A/mgₚₜ) > Pd₃Ru₁/Pt₃₋₄L NPs (0.93 A/mgₚₜ) > Pd₃Ru₁/Pt₅₋₆L NPs (0.38 A/mgₚₜ) > commercial Pt/C (0.18 A/mgₚₜ), whereas their SAs show the same order of Pd₃Ru₁/Pt₁₋₂L NPs (3.16 mA/cm²) > Pd₃Ru₁/Pt₃₋₄L NPs (1.19 mA/cm²) > Pd₃Ru₁/Pt₅₋₆L NPs

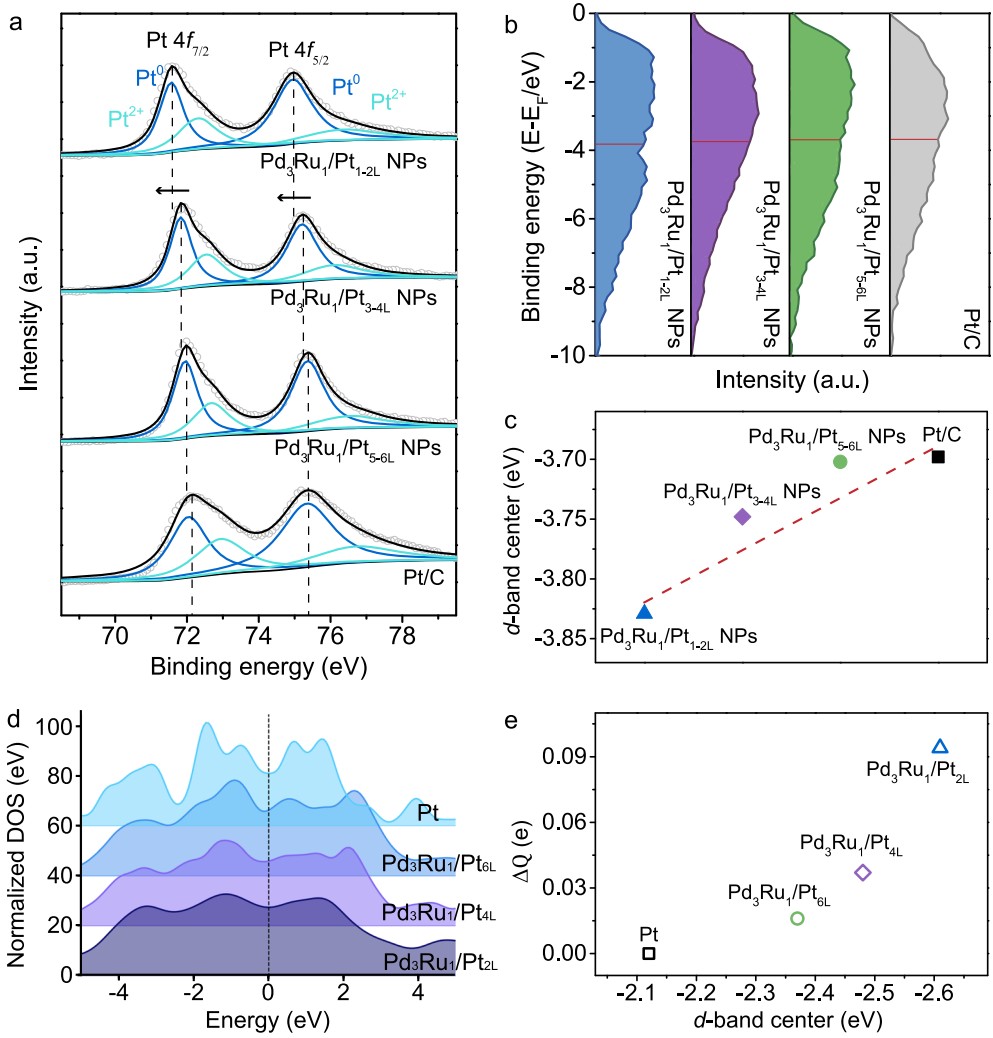

**Fig. 3 | XPS and DFT analyses of electronic structure of surface Pt. a** Pt 4f XPS spectra and **b** XPS valence band spectra measured for Pd₃Ru₁/Pt$_{1-2L}$ NPs, Pd₃Ru₁/Pt$_{3-4L}$ NPs, Pd₃Ru₁/Pt$_{5-6L}$ NPs and commercial Pt/C. **c** The *d*-band center shifts calculated from the integration of XPS valence band spectra in **b**. **d** The normalized DOS and **e** Bader charge and DFT calculated *d*-band center of the Pd₃Ru₁/Pt$_{2L}$ model, Pd₃Ru₁/Pt$_{4L}$ model, Pd₃Ru₁/Pt$_{6L}$ model, and pure Pt.

(0.87 mA/cm²) > commercial Pt/C (0.28 mA/cm²) (Supplementary Fig. 15b). The variation trend for ORR activities of Pd₃Ru₁/Pt$_{nL}$ NPs catalysts with exclusive ligand effect in the acidic electrolyte is consistent with the results under alkaline conditions. The optimal Pd₃Ru₁/Pt$_{1-2L}$ NPs also render a competitive ORR activity in acid electrolytes among reported state-of-the-art Pt-based core/shell catalysts (Supplementary Table 7). The durability tests (Supplementary Fig. 16) reveal that the Pd₃Ru₁/Pt$_{nL}$ NPs retain more than half of the initial values of MAs and ECSAs after 30,000 potential cycles, respectively (n = 1−2 L, 52.1% and 52.8%; n = 3−4 L, 58.1% and 59.7%; n = 5−6 L, 57.9% and 67.1%), higher than those of commercial Pt/C catalyst (27.8% and 38.4%) (Supplementary Fig. 17a, b). Worth mentioning that the ORR activity of Pd₃Ru₁ NPs was also examined in alkaline and acidic electrolytes, respectively (Supplementary Fig. 18). The Pd₃Ru₁ NPs are inert toward ORR, excluding the physical effect of Pd₃Ru₁ to the improvement of ORR catalytic performance.

**Correlation of ORR activities and *d*-band centers under exclusive ligand effect**
Sabatier principle has guided the advancing of heterogeneous catalysis for almost one century and well maps the activity of catalytic elements (especially transition metals) as a function of binding energy in a volcano manner[23]. Each time when the catalytic element of interest (Pt in

the case of ORR discussed here) is not satisfactorily efficient, it is customary to introduce another element to appropriately shift the catalytic element toward the peak *via* the so-called alloying strategy. Furthermore, configuring such alloys in various core-shell manners has proven benefits to the ORR electrocatalysis in terms of specific and mass activities, as well as stabilities. This improvement has been generally ascribed to the co-presence of ligand and strain effects that tune the oxygen adsorption toward the optimum. To clarify the origin, it is crucial to quantify how much each effect contributes to the activity improvement. One decade ago, Strasser and co-workers reported the first attempt to separate the strain effect from the convolution by using a dealloyed catalyst model[24]. With that model, a relation between ORR activity and the lattice strain of catalytic Pt could be experimentally established. However, to date, there is no reported experimental effort to separate and quantitate ligand effect in a real catalytic structure, although this effect is widely assumed to play a major role in promoting electrocatalysis over alloys. This study aims to fill this knowledge gap.

Experimentally, we avoided the generation of surface strain by designing an alloy composition (Pd₃Ru₁) with the same lattice parameter as pure Pt. Our presented microscopic and spectroscopic evidence (Figs. 1 and 2) strongly supported the key assumption of our work that the as-constructed core/shell structure (Pd₃Ru₁/Pt$_{nL}$)

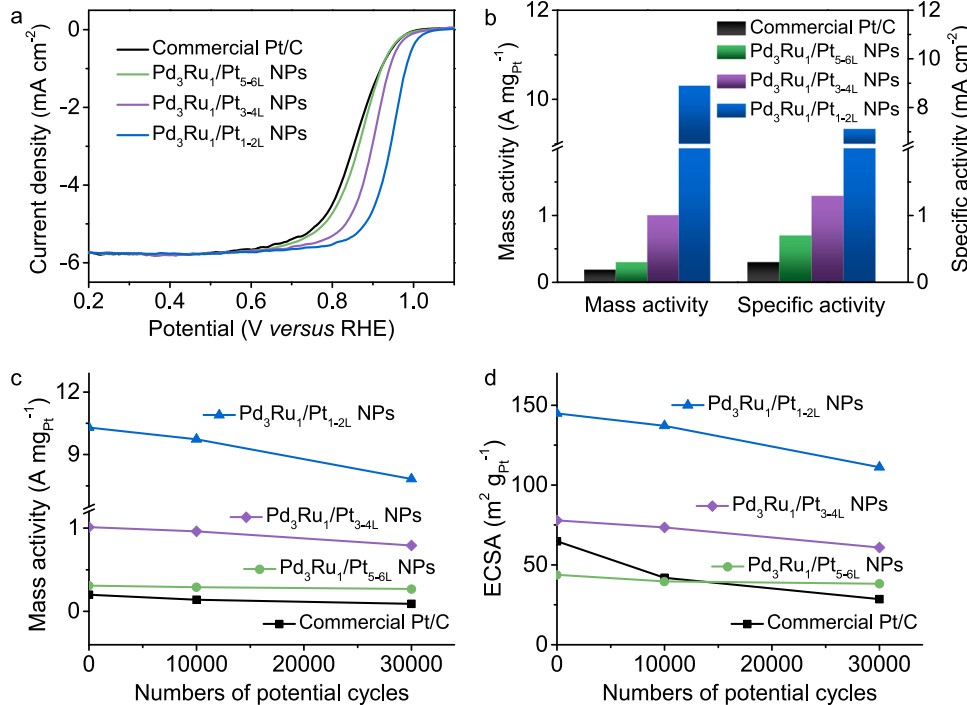

**Fig. 4 | Electrocatalytic performance of Pd₃Ru₁/PtₙL NPs and commercial Pt/C catalysts for ORR. a** ORR polarization curves were recorded in O₂-saturated 0.1 M KOH solution at a sweep rate of 20 mV/s and a rotation rate of 1600 rpm. **b** Column diagrams of MA and SA of different catalysts at 0.9 V *versus* RHE. **c** MA changes and **d** ECSA changes of different catalysts before and after 10,000 and 30,000 potential cycles between 0.6 V and 1.0 V *versus* RHE.

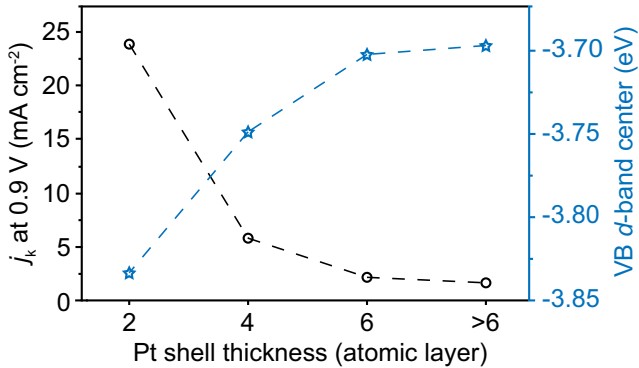

**Fig. 5 | Correlation of ORR activities and *d*-band centers of Pt shell with different thickness under exclusive ligand effect.** Differences of kinetic current density (*j*ₖ at 0.9 V) and *d*-band center calculated from valence-band spectra, respectively, between Pd₃Ru₁/PtₙL and commercial Pt/C catalysts.

generated no surface strain. On the other hand, studies on the electronic structures of Pd₃Ru₁/PtₙL by XPS and DFT (Fig. 3) confirmed the shift of the *d*-band center of Pt *via* the exclusive ligand effect. It was also evident that the extent of such exclusive ligand effect can be experimentally tuned by controlling the thickness of the Pt shell, which then motivated us to map ORR activities and *d*-band centers as a function of the number of atomic layers of the Pt shell, as shown in Fig. 5. We stress here that Fig. 5 is unprecedented because all involved parameters are experimentally measurable, and their variations are due to exclusive ligand effect. Most of the prior structure-performance relations were given based on either computational or ligand-and-strain convoluted electronic structure parameters. For the first time, our strategy enables the experimental visualization of exclusive ligand effect in heterogeneous catalysis and is expected to provoke future

efforts to re-examine the role of ligand effect (experimentally) in other electrocatalytic reactions relevant to renewable energy conversions.

## Discussion

In conclusion, we design and precisely synthesize a class of Pd₃Ru₁/PtₙL core/shell model catalyst with exclusive ligand effect for greatly boosting the electrocatalytic activity. We experimentally and theoretically demonstrate that the electron transfer from the Pd₃Ru₁ core to Pt can result in an increase of electron density in the 5*d* orbitals of Pt, thus the downshift of the *d*-band center of Pt, which can lead to greatly enhanced catalytic performance. The ligand-effect optimized Pd₃Ru₁/Pt₁₋₂L NPs show extraordinary half-wave potential of 0.945 V and MA of 10.3 A/mg$_{Pt}$ at 0.9 V *versus* reversible hydrogen electrode (*vs*. RHE) towards ORR in alkaline electrolyte, which is among the best ORR Pt-based catalysts. In acid electrolytes, the Pd₃Ru₁/Pt₁₋₂L NPs also show a superior MA of 4.59 A/mg$_{Pt}$ for ORR. They are stable for ORR in both alkaline/acid electrolytes by showing MA decay of 21.8%/47.9% after 30,000 potential cycles, less than those of commercial Pt/C catalysts (55.0%/72.2%). The present work provides the first experimental evidence for exclusive ligand effect on electrocatalysis and also reports a promising class of core/shell nanostructured electrocatalysts for fuel cells bottlenecked ORR.

## Methods

### Synthesis of Pd$_x$Ru$_{1-x}$ NPs

In a typical synthesis of Pd$_x$Ru$_{1-x}$ NPs, the designated amounts of Pd(acac)₂, Ru₃(CO)₁₂, and ascorbic acid were added to 5.0 mL oleyla-mine in a 15 mL high-pressure glass vial (Chongqing Synthware Glass, P160001). Then, the glass vial was sealed, and the mixture was stirred in an oil bath at 45 °C for 1 h to get a homogeneous solution. The solution was heated to 180 °C for 6 h until cooling to room temperature. The resulting black colloidal product was collected by centrifugation (at a speed of 8000*g* for 5 min), washed three times with cyclohexane and ethanol, and redispersed in 10 mL oleylamine for further use.

The amounts of corresponding reagents in the synthesis of the $Pd_{97}Ru_3$ NPs, $Pd_{85}Ru_{15}$ NPs, $Pd_{75}Ru_{25}$ NPs, and $Pd_{50}Ru_{50}$ NPs were summarized in Supplementary Table 1, respectively.

## Synthesis of $Pd_3Ru_1/Pt_{nL}$ NPs

Firstly, 3.3 mL $Pd_3Ru_1$ NPs oleylamine solution was dispersed in 15 mL benzyl alcohol and ultrasonicated for 30 min. Simultaneously, a Pt precursor solution was prepared by dissolving $Pt(acac)_2$ in the mixture of oleylamine and oleic acid. The as-prepared 18.3 mL solution of $Pd_3Ru_1$ NPs was injected into a 100 mL flask and heated at 180 °C with stirring for 15 min. Then, the Pt precursor solution was injected into the reaction flask at a rate of 5.00 mL h$^{-1}$. After the injection, the mixed solution was heated at 180 °C for 1 h with stirring. Subsequently, the reaction mixture was cooled down to room temperature and added to 50.0 mL of cyclohexane/ethanol (volume ratio: 2:1). Finally, the $Pd_3Ru_1/Pt_{nL}$ NPs were collected by centrifugation (at a speed of 8000g for 5 min) and washed twice with a mixture of cyclohexane/ethanol.

The amounts of $Pt(acac)_2$, oleylamine, and oleic acid in the synthesis of the $Pd_3Ru_1/Pt_{1-2L}$ NPs, $Pd_3Ru_1/Pt_{3-4L}$ NPs, and $Pd_3Ru_1/Pt_{5-6L}$ NPs were summarized in Supplementary Table 3, respectively.

## Characterization

TEM was conducted on HITACHI HT7700 at an accelerating voltage of 100 kV. High-angle annular dark-field scanning TEM (HAADF-STEM) and energy dispersive X-ray spectroscopy (EDS) element mappings were done on an aberration-corrected JEOL 2200FS STEM/TEM microscope at 300 kV equipped with a Bruker-AXS SDD detector. X-ray diffraction (XRD) patterns were collected on the Rigaku D/max-2500 powder diffractometer with Cu-Kα radiation ($\lambda = 0.15406$ nm). The specific elemental composition of $Pd_xRu_{1-x}$ NPs and $Pd_3Ru_1/Pt_{nL}$ NPs and the concentration of catalysts were determined by an inductively coupled plasma optical emission spectrometer (Agilent 5110 ICP-OES). The X-ray absorption fine-structure spectroscopy (XAFS) at Pt $L_3$-edge was performed at the 1W1B beamline of Beijing Synchrotron Radiation Facility (BSRF, 2.5 GeV, a maximum current of 250 mA, Si (311) double-crystal). All samples were tableted and measured at room temperature under fluorescence mode. The Athena module of the IFEFFIT software package was employed to analyze the acquired XAFS raw data according to the standard data analysis procedures, and the data fitting was performed using the Artemis in IFEFFIT[25]. The X-ray photoelectron spectroscopy (XPS) was conducted with the Thermo Scientific K-AlphaK spectrometer. Based on the XPS valence band spectra, the $d$-band center of gravity ($\varepsilon_d$) of the metal catalyst was calculated in the range of −10.0 to 0 eV by using the following Eq. (1).

$$\varepsilon_d = \frac{\int N(\varepsilon)\varepsilon d\varepsilon}{\int N(\varepsilon)d\varepsilon} \qquad (1)$$

## Electrochemical measurements

The $Pd_3Ru_1/Pt_{nL}$ NPs catalysts were prepared by depositing the as-synthesized $Pd_3Ru_1/Pt_{1-2L}$ NPs, $Pd_3Ru_1/Pt_{3-4L}$ NPs and $Pd_3Ru_1/Pt_{5-6L}$ NPs onto the carbon black (Vulcan XC-72), respectively. In a typical preparation, the $Pd_3Ru_1/Pt_{nL}$ NPs dispersed in cyclohexane were mixed with carbon black dispersed in ethanol under ambient sonication for 1 h. After stirring for another 6 h, the product was collected by centrifugation at the speed of 8000g for 5 min, and washed twice with ethanol, then dried at 70 °C for 3 h. Before the electrochemical tests, we did further thermal treatment of the obtained catalysts at 220 °C for 2 h in an $N_2$ atmosphere to clean up the organic residue on the nanocrystals. The metal (Pd + Ru + Pt) loading amount on carbon was

controlled to be about 20.0 wt%, and the actual loading was analyzed by ICP-OES.

The catalyst inks with a concentration of 1.00 $mg_{catalysts}$/mL were prepared by dispersing the catalyst powder into a mixture of iso-propanol, ultrapure water, and Nafion solution with a volume ratio of 1:1:0.0025 through sonication. The specific Pt loadings of commercial Pt/C ink, $Pd_3Ru_1/Pt_{5-6L}$ NPs ink, $Pd_3Ru_1/Pt_{3-4L}$ NPs ink, and $Pd_3Ru_1/Pt_{1-2L}$ NPs ink were 0.200, 0.147, 0.111, and 0.045 $mg_{Pt}$/mL, respectively. Then, 10 μL of catalyst inks was dropped on the surface of working electrodes for the electrochemical tests.

All electrochemical tests were carried out in a three-electrode system by using a rotating disk electrode device (Pine Research Instrumentation, USA) connected to a CHI750e electrochemical workstation (Shanghai Chenhua Instrument Corporation, China) at room temperature. In detail, a glassy carbon rotating disk electrode with a diameter of 5.0 mm was used as the working electrode, Pt foil (1.0 × 1.5 cm$^2$) was used as a counter electrode, and Hg/HgO electrode or saturated calomel electrode was used as a reference electrode for alkaline or acid electrolyte, respectively. The electrochemically active surface area (ECSA) of the catalyst was determined by integrating the hydrogen adsorption charge on the cyclic voltammetry (CV) with a scan rate of 50 mV/s at room temperature in $N_2$-saturated 0.1 M $HClO_4$ solution. To evaluate the ORR kinetics of the above-mentioned catalysts, the linear scan voltammetry (LSV) measurements were performed in $O_2$-saturated 0.1 M KOH/$HClO_4$ solution at a scan rate of 20 mV/s with a rotating rate of 1600 r/min. Besides, the ORR electrochemical durability tests were measured in $O_2$-saturated 0.1 M KOH/$HClO_4$ solution by conducting CV curves at the potential between 0.6 and 1.0 V *versus* RHE at a sweep rate of 50 mV/s for 10,000 and 30,000 potential cycles, respectively.

## DFT calculations

Our total energy calculations were performed within the framework of DFT using the projector augmented plane-wave (PAW) method, as implemented in the Vienna ab initio Simulation Package (VASP)[26]. The generalized gradient approximation (GGA) proposed by Perdew, Burke, and Ernzerhof (PBE) was selected for the exchange-correlation potential[27]. A long-range van der Waals interaction was described by the DFT-D3 approach[28]. The cut-off energy for the expanding plane wave was set to 450 eV. The energy criterion was treated as 10$^{-5}$ eV in the iterative solution of the Kohn–Sham equation. We built three configurations of the surface superstructure with respect to experimental observations: n-layer (n = 2, 4, 6) (2 × 2) Pt shell on $Pd_3Ru_1$ surface. The 2-layer pristine Pt was also built as a reference. To avoid artificial interaction between periodic images, a vacuum layer of 16 Å was added along the perpendicular z direction of the crystalline structure. The Brillouin zone integration was sampled using a Monkhorst–Pack (3 × 3 × 1) k-mesh. The atomic positions were fully relaxed until the residual forces on the per atom were less than 0.01 eV/Å. For DOS simulation, a (19 × 19 × 1) k-mesh was used. The normalized DOS was utilized to analyze the electron contributions in the $Pd_3Ru_1/Pt_{nL}$ configurations and the normalized DOS was defined as the total DOS of $Pd_3Ru_1/Pt_{nL}$ divided by the number of Pt layers, such as 2 layers, 4 layers, and 6 layers. The average charge transfer was determined by the following Eq. (2).

$$\Delta Q = \frac{1}{n(Q_a - Q_b)} \qquad (2)$$

$\Delta Q$ represents the difference in the Bader charge, $Q_a$ and $Q_b$ described the atomic Bader charge of $Pd_3Ru_1/Pt_{nL}$ and original states of the valence electron in pristine Pt, n was the number of Pt atoms in the corresponding $Pd_3Ru_1/Pt_{nL}$ model.

## Data availability
The data generated in this study are available within the paper and Supplementary Information. Source data are provided in this paper.

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

## Acknowledgements
S.G. acknowledge the fundings from National Key R&D Program of China (No. 2021YFA1501001), National Science Fund for Distinguished Young Scholars (No. 52025133), National Natural Science Foundation of China (No. 52261135633), the Beijing Natural Science Foundation (No. Z220020), NEW Cornerstone Science Foundation through the XPLORER PRIZE and CNPC Innovation Found (2021DQ02-1002). L.T. thanks the financial support from the China Postdoctoral Science Foundation (No. 2019M660290). The authors thank the BL1W1B station at Beijing Synchrotron Radiation Facility (BSRF) for XAFS measurement.

## Author contributions
S.G. conceived the project. L.T. designed the research. L.T., K.W., and H.G. performed the synthesis, general characterization, and electrochemical tests. L.T., F.Lv., H.M., F. Lin, and H.L. participated in the XAFS and XPS experiments and data analyses. L.T., Q.Z., and L.G. conducted HAADF-STEM measurements and data analyses. L.T and M.L. carried out the DFT analyses. All the authors discussed the results. S.G., L.T., and M.L. wrote the paper.

## Competing interests
The authors declare no competing interests.
