## [Peer Review File · Nature Communications]

Precise synthetic control of exclusive ligand effect boosts oxygen reduction catalysisREVIEWER COMMENTS

Reviewer #1 (Remarks to the Author):

Prof. Guo et al. described a synthetic strategy for Pd₃Ru₁/Pt core/shell nanoplates that shed insights into the relationship between ligand effect and electrocatalytic activity. The finding is interesting and can be insightful to electrocatalysis research. On the other hand, a few clarifications should be addressed. For questions, please refer to the attachment.

Reviewer #2 (Remarks to the Author):

This paper reports the preparation of Pd₃Ru₁/Pt core/shell nanoplates (NPs) for their use as electrocatalysts for the alkaline oxygen reduction reaction (ORR). Pd₃Ru₁/Pt NPs have well-matched core and shell lattice parameters, which allows for exclusively isolating ligand effect, which is commonly convoluted with strain effect. X-ray absorption near-edge structure and X-ray photoelectron spectroscopy results and density functional theory calculations suggested an electron transfer from the Pd₃Ru₁ core to Pt shell in the shell-thickness-controlled Pd₃Ru₁/Pt_nL NPs, resulting in the downshift of the d-band center of Pt. The optimal Pd₃Ru₁/Pt₁-2L NPs achieved a high mass activity of 10.3 A/mgPt at 0.90 V for the alkaline ORR and also exhibited good stability for the ORR.

This paper presents interesting results, which are a topic of continuous interest in the field but have not yet been established. However, several aspects of the paper should be clarified for further consideration of this paper in Nature Commun.

- 1) Title: The title is too concise. It should include more detailed information.
- 2) Literature: Many important previous works on the core/shell nanoparticle-based ORR catalysts, including those of Radosalv Adzic and Younan Xia, are missed from the references. Also, some important reviews presenting recent advances in Pt-based ORR catalysts are omitted.
- 3) For the alkaline measurements, the use of SCE is not recommended; OH⁻ can permeate through the electrode, which can invoke changes in the electrode potential. The Hg/HgO electrode should be used. This recommendation has been repeatedly suggested in recent literature.
- 4) Pt-based materials have been mainstays as acidic ORR electrocatalysts. The activity and durability of Pd₃Ru₁/Pt_nL catalysts should be examined in an acidic electrolyte. Their catalytic performance should be compared with those of state-of-the-art Pt-based catalysts.
- 5) The authors claimed throughout the manuscript that by synthesizing core/shell NPs having exclusively the ligand effect they could significantly boost the ORR activity, which is manifested as the best alkaline ORR activity. Then, the Pd₃Ru₁/Pt_nL catalysts have better activity than the catalysts having both ligand and strain effects? If this is demonstrated appropriately, their claim is valid. But if not, they should tone down.
- 6) The d-band center derived from the XPS band edge spectra (Figure 5b,c) suggests the downshift of the d-band center from Pt/C in the order of Pt/C (-3.698 eV) > Pd₃Ru₁/Pt₅-6L NPs (-3.702 eV) > Pd₃Ru₁/Pt₃-4L NPs (-3.748 eV) > Pd₃Ru₁/Pt₁-2L NPs (-3.829 eV). Yet, this trend does not match perfectly with XANES data, which suggested higher white line intensity (i.e. d-electron vacancy) for Pd₃Ru₁/Pt₅ than Pt. This inconsistency should be clarified.
- 7) Among composition-controlled PdRu NPs (Supplementary Fig. 1), Pd₅₀Ru₅₀ NPs have a crystal structure of fcc like other samples. Given Pd and Ru have fcc and hcp as the most stable phases in their bulk state, respectively, they can be stabilized in either fcc or hcp. Why

do Pd₅₀Ru₅₀ NPs have the fcc phase? With a higher content of Ru, the stabilized phase will change?

8) Page 13: "Our presented microscopic and spectroscopic evidence (Fig. 1 and Fig. 2) strongly supported the key assumption of our work that the as-constructed core/shell structure (Pd₃Ru₁/Pt_nL) generated no surface strain." For the validity of this claim, experimental proof should be provided. Geometric phase analysis can address this issue.

9) Curve fitting results for EXAFS spectra should be provided.

10) Electrochemically active surface areas of the catalysts should be provided.

11) Vendor name and product name should be provided for the commercial Pt/C catalyst used as the benchmark.

12) Supplementary Fig. 3: elemental analysis results for each element should also be given in wt%.

13) Page 5, lines 4-5: "Figs 1 and 2" to "Supplementary Fig. 1".

Reviewer #3 (Remarks to the Author):

The manuscript by Guo and coworkers reports on Pd-Ru/Pt electrocatalysts for the ORR. The authors should be praised for the systematic work on the synthesis and structural characterization of the nanoparticles presented in this work. However, the overall impact of this study should be justified by its novelty, and that would be challenging. There are many articles in the literature similar to this study, and it would be difficult to distinguish this work from already published reports. No new phenomena were observed, and measurements were done in alkaline electrolyte by RDE, which is additional obstacle to designate this study as the high impact report. All three metals that make this catalyst belong to the group of precious metals and evaluating this material in alkaline electrolyte does not come with appeal for practical applications.

The work is publishable, however, it belongs to the journal that is entirely focused on electrocatalysis, such as for example, *Electrochimica Acta*.

Reviewer #4 (Remarks to the Author):

In this manuscript, the authors reported a new kind of core/shell Pd₃Ru₁/Pt nanoplate catalyst with exclusive ligand effect for greatly boosting the electrocatalysis of oxygen reduction reaction (ORR). They demonstrate that the layer-by-layer growth of atomic Pt overlayers onto PdRu nanoplates can guarantee no lattice mismatch between core and shell, leading to exclusive ligand effect. The precise design of exclusive ligand effect for enhancing electrocatalysis is not reported before, and is significant in the fields of catalytic science and materials science, etc. I would like to recommend this nice work to be published in *Nat Commun* after addressing the following minor issues.

1. The authors reported a negative shift of Pt 4f binding energy in the Pd₃Ru₁/Pt_nL (from n:5-6L, to n:3-4L, to n:1-2L), and ascribed this shift to electron transfer from Pd₃Ru₁ to Pt shell. Please provide more discussion about Pd and Ru.

2. In the DFT calculation section, how to define the normalized DOS of different Pd₃Ru₁/Pt_nL configurations. Detailed description of DFT should be given.

3. According to the "Methods" section, oleylamine was utilized as solvent in the process of fabricating the crystals. How to clean up the organic residue on the obtained catalysts before the catalyst?

4. Supplementary Fig. 12 was not mentioned in the manuscript. The authors should provide the related discussion/explanation.

5. Detailed XAFS curve fitting parameters should be supplied.
6. There are some grammatical errors in the manuscript that need to be corrected. For instance, "Normalized with Pt loading." is not a complete sentence; "is not satisfactory efficient" should be "is not satisfactorily efficient".

Prof. Guo *et al.* described a synthetic strategy for Pd₃Ru₁/Pt core/shell nanoplates that shed insights into the relationship between ligand effect and electrocatalytic activity. The finding is interesting and can be insightful to electrocatalysis research. On the other hand, a few clarifications should be addressed.

In Page 7-8, XAS section

1. "...WL is regarded as a qualitative indicator of electron **vacancies** in the 5d orbitals of Pt atoms".
 - XANES reveals how the electron density in orbitals affects electron transitions. Although changes in white line are indicative, they are not essential for detecting electron vacancies in 5d orbitals. Therefore, I recommend using electron density instead.
2. "...indicating the significant decreased number of d-band hole,..."
 - It is hardly convincing to say "significant" with only around 0.2 normalized absorbance change in XANES. I suggest removing "significant".
3. "XANES depicts a lower threshold energy for the Pd₃Ru₁/Pt₁₋₂L NPs compared with that for Pt foil (inset of Fig. 2a), suggesting a relative lower valence state of Pt₁₋₂L".
 - For Pt L₃ edge, the estimation of valence state relies on the white line intensity rather than threshold energy (which is often seen in K-edge XANES analysis). I suggest removing this sentence because the interpretation is not accurate.
4. "there is no notable changes on Pt-Pt bond length of PtnL at around 2.51 Å"

In Figure 2, three peaks between 2.0 and 3.5 Å (without phase correction) are partially overlapping, but the authors only discussed the peak at 2.5 Å. The intensities of these peaks do not vary proportionally compared to those of Pt foil. The addition of heteroatoms such as Pd and Ru can coordinate with Pt, resulting in different wavefunctions in Pt XAS and leading to differences in relative peak intensities. Therefore, it would be inappropriate to select one peak and claim that there are "no notable changes." In my opinion, it is advisable to remove the EXAFS section to avoid this oversimplified explanation. Page 14, Method, Synthesis of nanoplates

Can the authors provide the full names for OAm and OA? These abbreviations are not defined in the context. Furthermore, using "X Y Z" to describe the chemical usage may be redundant since the details are provided in the supporting information. I suggest rephrasing it as "a designated amount of chemicals was used..." and "the chemical quantities for each nanoplate synthesis are available in the supporting information."

Page 15, Characterization

Could authors cite the publication for Athena software?

B. Ravel and M. Newville, ATHENA, ARTEMIS, HEPHAESTUS: data analysis for X-ray absorption spectroscopy using IFEFFIT, Journal of Synchrotron Radiation **12**, 537–541 (2005) [doi:10.1107/S0909049505012719](https://doi.org/10.1107/S0909049505012719)

Supporting Information, Page 15, Table 3

The data presented in Table 3 appears to be raw data from the ICP software, but providing the concentration of each component would be more straightforward for readers to understand. Additionally, I am interested in knowing whether the XAS measurements were conducted through transmission or fluorescence mode for these nanoplate materials. The metal concentration information could be helpful in understanding the measurement details.

To Reviewer 1:

General comments: Prof. Guo *et al.* described a synthetic strategy for Pd₃Ru₁/Pt core/shell nanoplates that shed insights into the relationship between ligand effect and electrocatalytical activity. The finding is interesting and can be insightful to electrocatalysis research. On the other hand, a few clarifications should be addressed.

Authors' Response: We thank the reviewer for appreciating our results are interesting. All of your comments are highly important and valuable for us to improve the manuscript. We addressed the comments point-by-point and made the corresponding changes accordingly in the revised manuscript. We sincerely hope that the revised manuscript will satisfy your stringent criteria for publication.

In Page 7-8, XAS section.

Q1-1: "...WL is regarded as a qualitative indicator of electron vacancies in the 5d orbitals of Pt atoms".

- XANES reveals how the electron density in orbitals affects electron transitions. Although changes in white line are indicative, they are not essential for detecting electron vacancies in 5d orbitals. Therefore, I recommend using electron density instead.

A1-1: Thanks for your kind comments. We replaced the statement of "electron vacancies" by "electron density" in the revised manuscript.

Q1-2: "...indicating the significant decreased number of d-band hole,..."

- It is hardly convincing to say "significant" with only around 0.2 normalized absorbance change in XANES. I suggest removing "significant".

A1-2: Thanks for your kind suggestion. We removed "significant" in the revised manuscript.

Q1-3: "XANES depicts a lower threshold energy for the Pd₃Ru₁/Pt_{1-2L} NPs compared with that for Pt foil (inset of Fig. 2a), suggesting a relative lower valence state of Pt_{1-2L}".

- For Pt L₃ edge, the estimation of valence state relies on the white line intensity rather than threshold energy (which is often seen in K-edge XANES analysis). I suggest removing this sentence because the interpretation is not accurate.

A1-3: Thanks for your kind suggestion. We removed the wrong interpretation.

Q1-4: "there is no notable changes on Pt-Pt bond length of PtnL at around 2.51 Å". In Figure 2, three peaks between 2.0 and 3.5 Å (without phase correction) are partially overlapping, but the authors only discussed the peak at 2.5 Å. The intensities of these peaks do not vary proportionally compared to those of Pt foil. The addition of heteroatoms such as Pd and Ru can coordinate with Pt, resulting in different wavefunctions in Pt XAS and leading to differences in relative peak intensities. Therefore, it would be inappropriate to select one peak and claim that there are "no notable changes." In my opinion, it is advisable to remove the EXAFS section to avoid this oversimplified explanation.

A1-4: Thanks for your valuable comments. We removed the oversimplified descriptions. And also we restudied the electronic structure of Pt and the electronic interaction between Pd₃Ru₁ core and Pt shell by XAFS in the revised manuscript. Specifically, the XAFS characterizations of all samples were retested (revised Fig. 2), the data were fitted (Supplementary Fig. 8 and Table 4), and the corresponding results were further discussed in the XAS section.

Q2: Page 14, Method, Synthesis of nanoplates

Can the authors provide the full names for OAm and OA? These abbreviations are not defined in the context. Furthermore, using "X Y Z" to describe the chemical usage may be redundant since the details are provided in the supporting information. I suggest rephrasing it as "a designated amount of chemicals was used..." and "the chemical quantities for each nanoplate synthesis are available in the supporting information. "

A2: Thanks for your kind suggestion. We supplemented the full names for "OAm" and "OA" in the revised manuscript. Furthermore, we removed the wording of "X Y Z", and rephased it as "the designated amounts of ..." and "The additive amounts of corresponding reagents in the fabrication of ... were summarized in Supplementary Table 3, respectively." in the revised manuscript.

Q3: Page 15, Characterization

Could authors cite the publication for Athena software?

B. Ravel and M. Newville, ATHENA, ARTEMIS, HEPHAESTUS: data analysis for X-ray absorption spectroscopy using IFEFFIT, Journal of Synchrotron Radiation 12, 537–541 (2005) doi:10.1107/S0909049505012719.

A3: Thanks for your kind suggestion. We cited this publication in the revised manuscript.

Q4: Supporting Information, Page 15, Table 3

The data presented in Table 3 appears to be raw data from the ICP software, but providing the concentration of each component would be more straightforward for readers to understand. Additionally, I am interested in knowing whether the XAS measurements were conducted through transmission or fluorescence mode for these nanoplate materials. The metal concentration information could be helpful in understanding the measurement details.

A4: Thanks for your valuable suggestion. We provided the concentration of each component of catalysts with unit of "wt%" in the revised Supplementary Table 2. Additionally, the XAS experiments of all samples were conducted in a fluorescence mode, and the corresponding informations were supplemented in the "Characterization" section.

To Reviewer 2:

General comments: This paper reports the preparation of Pd₃Ru₁/Pt core/shell nanoplates (NPs) for their use as electrocatalysts for the alkaline oxygen reduction reaction (ORR). Pd₃Ru₁/Pt NPs have well-matched core and shell lattice parameters, which allows for exclusively isolating ligand effect, which is commonly convoluted with strain effect. X-ray absorption near-edge structure and X-ray photoelectron spectroscopy results and density functional theory calculations suggested an electron transfer from the Pd₃Ru₁ core to Pt shell in the shell-thickness-controlled Pd₃Ru₁/Pt_{nL} NPs, resulting in the downshift of the d-band center of Pt. The optimal Pd₃Ru₁/Pt_{1-2L} NPs achieved a high mass activity of 10.3 A/mg_{Pt} at 0.90 V for the alkaline ORR and also exhibited good stability for the ORR.

This paper presents interesting results, which are a topic of continuous interest in the field but have not yet been established. However, several aspects of the paper should be clarified for further consideration of this paper in Nature Commun.

Authors' Response: Thanks for your great efforts in reviewing our manuscript. We specially appreciate your valuable comments and suggestions. We have performed all the experiments suggested by you, further addressed the comments point-by-point and made the corresponding changes accordingly in the revised manuscript.

Q1: Title: The title is too concise. It should include more detailed information.

A1: Thanks for your kind suggestion. We amended the title as “Precise synthetic control of exclusive ligand effect boosts oxygen reduction catalysis”.

Q2: Literature: Many important previous works on the core/shell nanoparticle-based ORR catalysts, including those of Radosalv Adzic and Younan Xia, are missed from the references. Also, some important reviews presenting recent advances in Pt-based ORR catalysts are omitted.

A2: Thanks for your valuable comments. Based on your recommendation, we updated the citations in the revised manuscript.

Q3: For the alkaline measurements, the use of SCE is not recommended; OH⁻ can permeate through the electrode, which can invoke changes in the electrode potential. The Hg/HgO electrode should be used. This recommendation has been repeatedly suggested in recent literature.

A3: Thanks for your professional comments. In the revised electrochemical section, we chose Hg/HgO electrode or saturated calomel electrode as reference electrode for alkaline or acid measurements, respectively. And the corresponding test results were provided in the revised manuscript.

Q4: Pt-based materials have been mainstays as acidic ORR electrocatalysts. The activity and durability of Pd₃Ru₁/Pt_{nL} catalysts should be examined in an acidic electrolyte. Their catalytic performance should be compared with those of state-of-the-art Pt-based catalysts.

A4: Thanks for your valuable comments. The ORR catalytic performances of Pd₃Ru₁/Pt_{nL} catalysts were examined in the acidic electrolyte, and compared with that of state-of-the-art Pt-based core/shell catalysts (Supplementary Table 7) with a further discussion in the revised manuscript.

Q5: The authors claimed throughout the manuscript that by synthesizing core/shell NPs having exclusively the ligand effect they could significantly boost the ORR activity, which is manifested as the best alkaline ORR activity. Then, the Pd₃Ru₁/Pt_{nL} catalysts have better activity than the catalysts having both ligand and strain effects? If this is demonstrated appropriately, their claim is valid. But if not, they should tone down.

A5: Thanks for your valuable comment. With reference to all reported catalysts that having both ligand and strain effects, our synthesized Pd₃Ru₁/Pt_{nL} catalysts possessed excellent ORR catalytic performance but not the best in the acid electrolyte. We agree with your suggestion and deleted such descriptions of “the best ...” in the revised manuscript.

Q6: The d-band center derived from the XPS band edge spectra (Figure 5b,c) suggests the downshift of the d-band center from Pt/C in the order of Pt/C (-3.698 eV) > Pd₃Ru₁/Pt_{5-6L} NPs (-3.702 eV) > Pd₃Ru₁/Pt_{3-4L} NPs (-3.748 eV) > Pd₃Ru₁/Pt_{1-2L} NPs (-3.829 eV). *Yet*, this trend does not match perfectly with XANES data, which suggested higher white line intensity (i.e. d-electron vacancy) for Pd₃Ru₁/Pt₅ than Pt. This inconsistency should be clarified.

A6: Thanks for your professional comments. This inconsistency might be ascribed to our omissive but requisite normalized process and data fitting in the XAFS section. Based on the comments for XAFS section from all reviewers, we removed the oversimplified explanation and unnormalized experimental results, instead, we restudied the electronic structure of Pt and the electronic interaction between Pd₃Ru₁ core and Pt shell by XAFS in the revised manuscript. Specifically, the XAFS characterization of all samples were retested (revised Fig. 2), the data were fitted (Supplementary Fig. 8 and Table 4), and the corresponding results were further discussed in the XAFS section.

Q7: Among composition-controlled PdRu NPs (Supplementary Fig. 1), Pd₅₀Ru₅₀ NPs have a crystal structure of *fcc* like other samples. Given Pd and Ru have *fcc* and *hcp* as the most stable phases in their bulk state, respectively, they can be stabilized in either *fcc* or *hcp*. Why do Pd₅₀Ru₅₀ NPs have the *fcc* phase? With a higher content of Ru, the stabilized phase will change?

A7: Thanks for your valuable comments. The crystal structures of various Pd_xRu_{100-x} NPs were further investigated by XRD. Learnt from the XRD results in the Figure R1, the dominant structure of the as-prepared Pd_xRu_{1-x} NPs changes from *fcc* to *hcp* with increasing Ru content. When the Ru content X increases greater than 50, the as-prepared Pd_xRu_{1-x} NPs gradually shows the coexistence of *fcc* and *hcp* structures.

Figure R1. PXRD patterns of different $\text{Pd}_x\text{Ru}_{100-x}$ NPs. PDF cards: Pt 04-0802; Pd 87-0637; Ru 06-0663.

Q8: Page 13: “Our presented microscopic and spectroscopic evidence (Fig. 1 and Fig. 2) strongly supported the key assumption of our work that the as-constructed core/shell structure ($\text{Pd}_3\text{Ru}_1/\text{Pt}_{\text{nL}}$) generated no surface strain.” For the validity of this claim, experimental proof should be provided. Geometric phase analysis can address this issue.

A8: Thanks for your valuable comment. We performed geometric phase analysis (GPA) on the $\text{Pd}_3\text{Ru}_1/\text{Pt}_{\text{nL}}$ NPs, and supplemented the corresponding false-colored GPA maps in the Supplementary Fig. 7.

The GPA results was discussed as follows in the revised manuscript. “Furthermore, the geometric phase analysis (GPA) on the random $\text{Pd}_3\text{Ru}_1/\text{Pt}_{1-6\text{L}}$ NPs were performed to recheck strain on the Pt shells. The corresponding false-colored GPA map show mostly uniform color throughout the NP, reconfirming a negligible in-plane strain (ϵ_{xx}) on the Pt shells (Supplementary Fig. 7).”

Q9: Curve fitting results for EXAFS spectra should be provided.

A9: Thanks for your valuable comments. The curve fitting results for EXAFS spectra were supplementd in the revised Supplementary Fig. 8 and Table 4.

Q10: Electrochemically active surface areas of the catalysts should be provided.

A10: Thanks for your kind suggestion. We provided the electrochemical active surface ares of catalysts in the Supplementary Table 5.

Q11: Vendor name and product name should be provided for the commercial Pt/C catalyst used as the benchmark.

A11: Thanks for your valuable comment. We supplemented the vendor name and product name of commercial Pt/C catalyst in the revised supplementary informance.

Q12: Supplementary Table 3: elemental analysis results for each element should also be given in wt%.

A12: Thanks for your kind suggestion. We provided the elemental analysis results with unit of “wt%” in the Supplementary Table 2.

Q13: Page 5, lines 4-5: “Figs 1 and 2” to “Supplementary Fig. 1”.

A13: Thanks for your careful reading. We replaced “Figs 1 and 2” by “Supplementary Figs. 1 and 2”.

To Reviewer 3:

General comments: The manuscript by Guo and coworkers reports on Pd-Ru/Pt electrocatalysts for the ORR. The authors should be praised for the systematic work on the synthesis and structural characterization of the nanoparticles presented in this work.

- However, the overall impact of this study should be justified by its novelty, and that would be challenging. There are many articles in the literature similar to this study, and it would be difficult to distinguish this work from already published reports.
- No new phenomena were observed, and measurements were done in alkaline electrolyte by RDE, which is additional obstacle to designate this study as the high impact report. All three metals that make this catalyst belong to the group of precious metals and evaluating this material in alkaline electrolyte does not come with appeal for practical applications.

The work is publishable, however, it belongs to the journal that is entirely focused on electrocatalysis, such as for example, *Electrochimica Acta*.

Authors' Response: Thanks for your great efforts in reviewing our manuscript. We specially appreciate your professional comments.

- Based on the comments and suggestions from all the reviewers, we further emphasized the innovation and scientific breakthrough in details in the revised manuscript. And the novelty of this work has become more distinct, as illustrated in the following:

The core/shell structural engineering achieved by the deposition of a few Pt atomic layers, on a heterogeneous substrate, has been regarded as an effective strategy to optimize their catalytic performance via both/either strain and/or ligand effects (ref., *Science*. 2016, 354, 1410). To clarify the origin, it is crucial to quantitate how much each effect contributes to the activity improvement.

One decade ago, researchers successfully separated strain effect from the convolution strain-ligand effects by using a dealloyed catalyst model. However, to the best of our knowledge, there is no experimental design can deconvolute and quantitate ligand effect in a real catalytic structure. This study aims to fill this knowledge gap.

The most critical challenge for this knowledge gap lies on that the Pt shell and heterogeneous substrates always differ in lattice parameter in a core/shell structure, thus giving rise to the inevitable strain effect. We rationally designed and precisely constructed a core/shell catalyst model with the same lattice parameter in the core and shell, which enables us to experimentally quantitate the ligand effect to ORR electrocatalysis for the first time.

- To date, there is no reported experimental effort to separate and quantitate ligand effect in a real catalytic structure, although this effect is widely assumed to play a major role in promoting electrocatalysis over alloys. This study provides the first experimental evidence for exclusive ligand effect on electrocatalysis, and the relationship of ORR activities and *d*-band centers as a function of the number of atomic layers of Pt shell was mapped in this work, which is unprecedented because all involved parameters are experimentally measurable, and their variations are due to exclusive ligand effect.

Additionally, the ORR catalytic performance of Pd₃Ru₁/Pt_{nL} catalysts were examined in the acidic electrolyte, and the comparison with that of state-of-the-art Pt-based core/shell catalysts were supplemented in the Supplementary Table 7 with a further discussion in the revised manuscript.

We emphasized the innovation and scientific breakthrough of this study in the revised manuscript and carefully addressed the comments from all reviewers. We sincerely hope that the revised manuscript will satisfy your stringent criteria for the publication of *Nature Communications*.

To Reviewer 4:

General Comments: In this manuscript, the authors reported a new kind of core/shell Pd₃Ru₁/Pt nanoplate catalyst with exclusive ligand effect for greatly boosting the electrocatalysis of oxygen reduction reaction (ORR). They demonstrate that the layer-by-layer growth of atomic Pt overlayers onto PdRu nanoplates can guarantee no lattice mismatch between core and shell, leading to exclusive ligand effect. The precise design of exclusive ligand effect for enhancing electrocatalysis is not reported before, and is significant in the fields of catalytic science and materials science, etc. I would like to recommend this nice work to be published in Nat Commun after addressing the following minor issues.

Authors' Response: Thanks for your great efforts in reviewing our manuscript. We are very grateful for your interest and appreciation of this work as well as the confirmation of its significance. All the comments are very valuable for us to improve this work, which has been carefully addressed in the revised manuscript. We sincerely hope that the revised manuscript will satisfy your stringent criteria for publication.

Q1: The authors reported a negative shift of Pt 4f binding energy in the Pd₃Ru₁/Pt_nL (from n:5-6L, to n:3-4L, to n:1-2L), and ascribed this shift to electron transfer from Pd₃Ru₁ to Pt shell. Please provide more discussion about Pd and Ru.

A1: Thanks for your kind suggestion. We supplemented the Pd 3d and Ru 3p XPS spectra of Pd₃Ru₁/Pt_nL NPs in the Supplementary Fig. 9, and provided further discussion about Pd and Ru binding energy shifts of Pd₃Ru₁/Pt_nL NPs in the revised manuscript.

Q2: In the DFT calculation section, how to define the normalized DOS of different Pd₃Ru₁/Pt_nL configurations. Detailed description of DFT should be given.

A2: Thanks for your valuable comment. In this work, the normalized DOS was defined as that total DOS of Pd₃Ru₁/Pt_nL divided by the number of Pt layers, such as 2 layers, 4 layers and 6 layers. We supplemented the detailed description of DFT in the revised manuscript.

Q3: According to the “Methods” section, oleylamine was utilized as solvent in the process of fabricating the crystals. How to clean up the organic residue on the obtained catalysts before the catalyst?

A3: Thanks for your valuable comment. We supplemented the cleanup details in the “Electrochemical measurements” in the revised manuscript as follows: “Before the electrochemical tests, the obtained catalysts should be received a further thermal treatment at 220 °C for 2 h in N₂ atmosphere to clean up the organic residue on the nanocrystals.”

Q4: Supplementary Fig. 12 was not mentioned in the manuscript. The authors should provide the related discussion/explanation.

A4: Thanks for your careful reading. We supplemented the discussion of Supplementary Fig. 12 in the revised manuscript as follows: “Worth mentioning that the ORR activity of Pd₃Ru₁ NPs was

also examined in alkaline and acidic electrolytes, respectively (Supplementary Figs. 18). The Pd₃Ru₁ NPs is inert toward ORR, excluding physical effect of Pd₃Ru₁ to the improvement of ORR catalytic performance of the Pd₃Ru₁/Pt_{nL} configuration.”

Q5: Detailed XAFS curve fitting parameters should be supplied.

A5: Thanks for your valuable comment. The detailed XAFS curve fitting parameters were supplemented in the revised Supplementary Fig. 8 and Table 4, and the corresponding results were further discussed in the XAFS section.

Q6: There are some grammatical errors in the manuscript that need to be corrected. For instance, “Normalized with Pt loading.” is not a complete sentence; “is not satisfactory efficient” should be “is not satisfactorily efficient”.

A6: Thanks for your careful reading. We corrected the wrong sentence as “Normalized with Pt loadings, the mass activities (MA) of different catalysts at 0.9 V were calculated to follow by the order of...”, and replaced “satisfactory” with “satisfactorily”. We carefully checked and improved the English writing in the revised manuscript.

REVIEWERS' COMMENTS

Reviewer #1 (Remarks to the Author):

The authors have clearly answered the questions from reviewers. Good work.

Reviewer #2 (Remarks to the Author):

The authors have adequately revised the manuscript by taking account into the reviewers' comments and concerns. The revised manuscript now appears suitable for publication in Nature Commun.

Reviewer #4 (Remarks to the Author):

My comments have been addressed